# Deconvolution of conformational exchange from Raman spectra of aqueous RNA nucleosides

Alex L. Wilson [1], Carlos Outeiral [1], Sarah E. Dowd [1], Andrew J. Doig [2], Paul L. A. Popelier [1], Jonathan P. Waltho [1,3] & Andrew Almond [1✉]

Ribonucleic acids (RNAs) are key to the central dogma of molecular biology. While Raman spectroscopy holds great potential for studying RNA conformational dynamics, current computational Raman prediction and assignment methods are limited in terms of system size and inclusion of conformational exchange. Here, a framework is presented that predicts Raman spectra using mixtures of sub-spectra corresponding to major conformers calculated using classical and ab initio molecular dynamics. Experimental optimization allowed purines and pyrimidines to be characterized as predominantly *syn* and *anti*, respectively, and ribose into exchange between equivalent south and north populations. These measurements are in excellent agreement with Raman spectroscopy of ribonucleosides, and previous experimental and computational results. This framework provides a measure of ribonucleoside solution populations and conformational exchange in RNA subunits. It complements other experimental techniques and could be extended to other molecules, such as proteins and carbohydrates, enabling biological insights and providing a new analytical tool.

[1] Manchester Institute of Biotechnology and Department of Chemistry, School of Natural Science, Faculty of Science and Engineering, The University of Manchester, M1 7DN Manchester, UK. [2] Division of Neuroscience and Experimental Psychology, Michael Smith Building, School of Biological Sciences, Faculty of Biology, Medicine and Health, The University of Manchester, M13 9PT Manchester, UK. [3] Krebs Institute for Biomolecular Research, Department of Molecular Biology and Biotechnology, The University of Sheffield, S10 2TN Sheffield, UK. ✉email: andrew.almond@manchester.ac.uk

Ribonucleic acids (RNAs) are key players in the central dogma of molecular biology; genetic information is transported, translated, and regulated by RNA. Their importance was underlined by the discovery of self-replicating RNA ribozymes[1], supporting the RNA World hypothesis of the early evolutionary history of life[2]. Molecular descriptions, deriving from three-dimensional (3D)-structural information, are required to understand the enigmatic mechanisms behind these diverse biological functions[3]. Being predominantly single stranded, RNAs are intrinsically more flexible than globular proteins and only super-atomic crystallographic resolutions (2.5–4 Å) are typically attainable[4]; ribose ring densities are often diffuse and ambiguous, indicating conformational degeneracy is present even in crystals[5,6]. Macromolecular complexes can alternatively be studied with cryogenic electron microscopy but again atomic resolution is typically not achievable[7]. Solution-phase nuclear magnetic resonance (NMR) has also been instrumental in the determination of functional, conformational, and structural information about RNA. However, while NMR can provide atomic resolution, it is not suited to systems beyond RNA oligonucleotides as they require high concentrations, exhibit low chemical shift dispersion, and necessitate complex and expensive isotopic labeling[8–10]. Established experimental techniques are therefore limited in their ability to achieve atomic resolution on RNA. Lack of resolution masks functionally-important ribose ring puckering and conformational exchange. For example, constituent ribose rings show a general preference for c3′-endo (north) puckering in helical polynucleotides[6], and the c2′-endo (south) sugar pucker in non-helical polynucleotides. Dynamical interconversion is therefore responsible for important macromolecular changes between A and Z RNA and between A, B, and Z DNA[11,12]. This is reinforced by the fact that locking the sugars into particular puckers perturbs the delicate hybridization process and provides a novel route to therapeutics, highlighting the functional importance of ribose conformational exchange[13]. Clearly, further experimental techniques are required to provide atomic resolution of backbone conformational preferences in RNA.

Both infrared and Raman spectroscopy are able to study solution phase RNA samples[14,15], without restrictions on molecular size, and potentially provide an inexpensive, rapid route to the elucidation of RNA conformation. However, deciphering the spectral complexity associated with large RNAs and assigning the constituent bands is a problem limiting the utility of Raman spectroscopy[16]. Computational prediction of vibrational spectra is an excellent route to assignment of Raman spectra, potentially expanding Raman's domain of applicability to routine 3D-structural determination, but computational tractability limits current methods, both in terms of realistic system size and inclusion of conformational exchange[17–20]. However, there have been recent successful attempts to simulate on the order of $10^3$–$10^4$ atoms using combined classical and quantum methods[21–28]. This type of scalable computational approach is required to accurately predict vibrational spectra of biological molecules, and include the contribution of physical effects such as conformational exchange. Consequently, there have been a number of recent attempts to derive spectra from ab initio simulations, but the contributions of slow conformational exchange to computed spectra are inadequately reproduced due to limits on feasible simulation timeframes[29].

Here, a Raman spectral prediction framework is applied to aqueous RNA nucleosides to distil spectral complexity into contributions of multiple conformers in a manner computationally scalable to much larger systems, by combining classical and ab initio molecular dynamics with experiment. Specifically, long-timescale classical simulations fully explore molecular phase space, and highly populous regions are further sampled with multiple short-timescale ab initio simulations, providing accurate dynamics around specific conformers. Autocorrelation of the molecular polarizability then allows for individual conformer spectral prediction and subsequent combination into an overall Raman spectrum. In this way, we achieve experimental agreement, opening up Raman spectroscopy as a simple, inexpensive experimental route to 3D-structure determination of nucleic acids to complement existing techniques. While we demonstrate this for the first time on RNA ribonucleoside building blocks, it may have fundamental applicability to a wide range of other molecules.

## Results and discussion

**Classical molecular dynamics ensemble generation.** An ensemble of 3D-conformations for each nucleoside was initially created to provide a Boltzmann-weighted distribution in order to predict the most populated conformers and the effect of RNA conformational exchange. Classical molecular dynamics (MD) simulations (5 μs) with explicit water molecules were used for this purpose. The resultant conformational space was divided into two dimensions corresponding to the major degrees of freedom: glycosidic torsional angle (χ) and ribose ring pucker (P). For χ, syn spanned −90° to +90°, whilst anti described the remaining 180°, using the standard angle definitions: O4′-C1′-N9/N1-C4/C2 for purines/pyrimidines, respectively. For P, south (inclusive of the c2′-endo conformer) spanned +90° to +270°, whilst north (inclusive of the c3′-endo conformer) described the remaining 180° (Fig. 1), using the method of Altona[30]. The resulting populations of each conformer were determined, with syn and anti dominating in the purines and pyrimidines, respectively (Fig. 2). All nucleosides, with the exception of guanosine, had a slight preference for the south ring conformer (~60%). Guanosine was observed to have a marked preference for the north conformer (~70%). The exchange rate between conformations was in the range 0.1–3.3 ns⁻¹

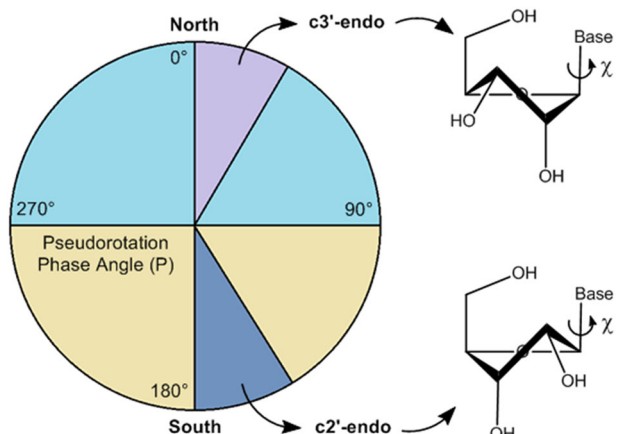

**Fig. 1 Ribonucleoside conformation.** The pseudorotational phase angle, referred to as ring pucker (P), is a simple description of the out-of-plane distortion of the ribose ring atoms and is one of the two major degrees of freedom in RNA ribonucleosides. When 90° < P < 270°, the ring is in a "south" conformation (pale yellow), whereas when P is outside of this range, the ring is in a "north" conformation (pale blue). In many RNA oligonucleotides, the ribose ring is typically observed with 144° < P < 180° (c2′-endo, blue), or 0° < P < 36° (c3′-endo, violet). For all four nucleosides studied in the present work, classical molecular dynamics simulations show a strong preference for the south conformer. The other major degree of freedom is the torsional angle between the ribose ring and the base, referred to as the glycosidic bond (χ). When −90° < χ < 90°, the base is in the syn-conformation, otherwise it is in the anti-conformation. Here, the combinations of the ribose ring with four different bases are studied: adenosine, guanosine, cytidine, and uridine.

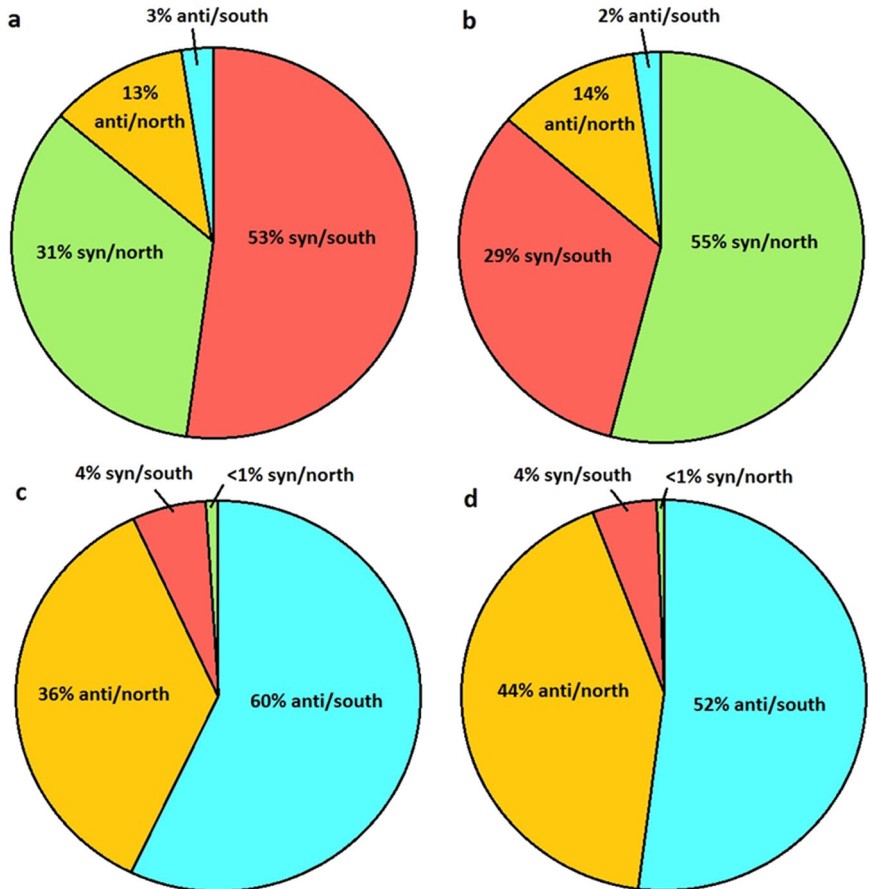

**Fig. 2 Conformer distributions from classical simulations. Conformer distributions from 5 μs classical molecular dynamics (MD) simulations in water, using a force-field optimized for RNA (OPLS-AA/M). a** Adenosine, **b** guanosine, **c** cytidine, and **d** uridine. For the purine nucleosides, the *syn*/south (red) and *syn*/north (green) conformers are found to dominate. For the pyrimidine nucleosides, the *anti*/south (blue) and *anti*/north (orange) conformers dominate.

(Supplementary Figs. 1 and 2), putting the mean lifetimes of conformers in the hundreds of picoseconds to tens of nanoseconds range, in agreement with fast exchange observed in previous NMR studies[31]. In the pyrimidines *anti* to *syn* transitions had a slower rate (~0.1 ns$^{-1}$), consistent with the preference for the *anti* conformer, while in purines it was the *syn* to *anti* transition that had a slower rate (~0.4 ns$^{-1}$).

**Raman spectral prediction from classical molecular dynamics.**
The Raman spectrum of each nucleoside was initially approximated by a single 3D-structure selected from the most populous region of its conformer distribution (Fig. 2). Prior to the determination of the Raman-active vibrational modes, each nucleoside was quantum-mechanically energy-minimized; a process that scales cubically with system size[32]. The resulting discrete frequencies for Raman-active vibrations of these single conformers were broadened using a Lorentzian function to 5 cm$^{-1}$ at full-width half-maximum (FWHM), thereby accounting for intrinsic peak lifetime[33]. Frequency scaling-factors were optimized to achieve experimental agreement[34]. Band assignment of characteristic experimental peaks was performed for each nucleoside using these single, most-populous conformers (Supplementary Table 1). In all cases, this approach produced highly-resolved spectra with fine structure, but in general had a poor agreement with experiment (Fig. 3), especially in terms of the relative intensities of characteristic bands. For example, for adenosine, the intense triplet observed experimentally (1300–1400 cm$^{-1}$) was not

predicted theoretically. Of particular note, experimental spectra were on average ~15 cm$^{-1}$ broader relative to predictions (Fig. 3). This implies that spectral linewidth is not solely due to intrinsic linewidth; non-uniform shape and width of experimental bands across all nucleosides points to this additional broadening being a result of conformational exchange[35]. Therefore, it is apparent that predictions based on the single most populated conformer (Fig. 3) and any of the other canonical conformers (Supplementary Fig. 3) may be insufficient to interpret and decipher the complex Raman spectra of macromolecular systems, such as RNA. Moreover, poor scaling of this approach with respect to system size represents a potential bottleneck that limits its applicability. However, if the effect of conformational exchange could be taken into account theoretically, while achieving scalability, the opportunity exists to extract novel information on molecular dynamics from the experimental Raman spectra.

The importance of including conformational exchange in Raman spectral predictions was confirmed by sampling 1000 3D-structures from each Boltzmann-weighted MD conformer distribution using aqueous molecular dynamics simulations and a force-field optimized for RNA (OPLS-AA/M). Each 3D-structure was geometrically optimized prior to Raman spectral calculation using B3LYP density functional theory (DFT), as above, and subsequent averaging over all 1000 generated spectra for each nucleoside was carried out to give single overall Raman spectra. These overall spectra were again frequency scaled, and broadened to 5 cm$^{-1}$ FWHM with a Lorentzian function. Inclusion of multiple conformers manifested as unresolved frequency and

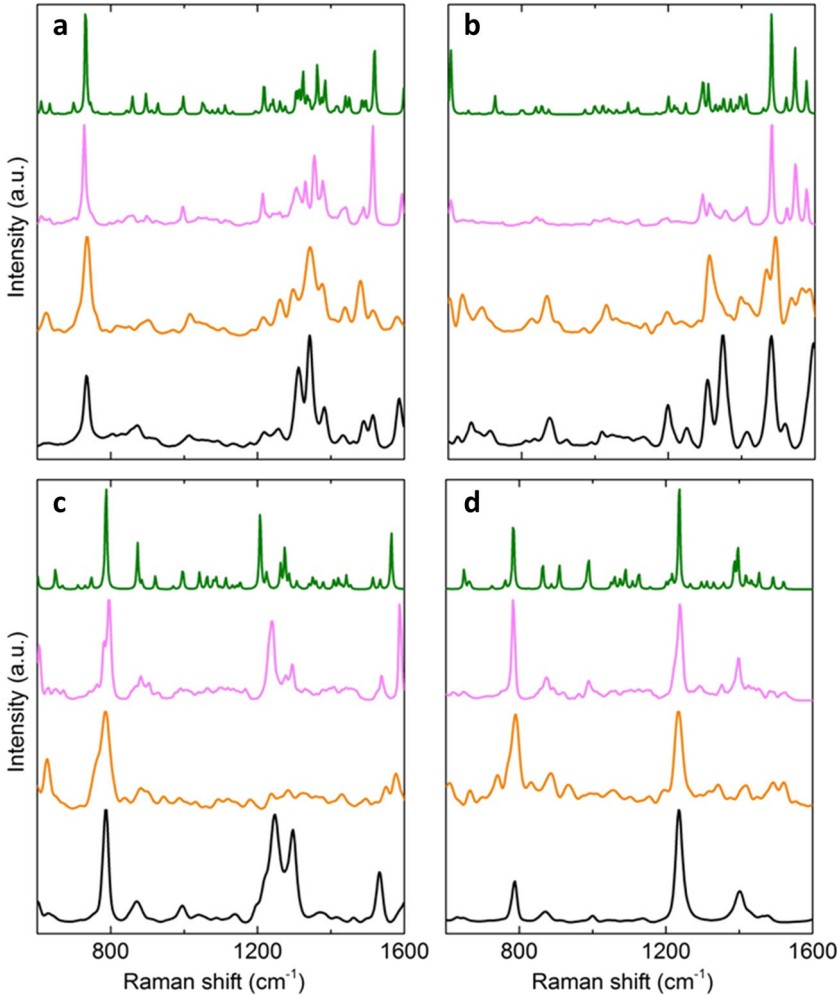

**Fig. 3 Comparison of different approaches to theoretically reproduce the experimental Raman spectra. a** Adenosine, **b** guanosine, **c** cytidine, and **d** uridine. The experimental Raman spectra are shown at the bottom of each panel (black). The Raman spectrum determined from a single QM-optimized conformer, from the most populous region (Fig. 2) of each nucleoside conformer distribution (green), resulted in spectra with no conformational exchange and therefore very little broadening. Attempting to account for conformational exchange by averaging over the individual spectra from 1000 conformers per nucleoside (pink) resulted in broader spectra, but relative peak intensities were not well reproduced. Conformers were extracted from classical molecular dynamics (MD) simulations in water, using a force-field optimized for RNA (OPLS-AA/M). Inclusion of local dynamics via ab initio simulation of the *syn*/south conformer region led to spectra (orange) with improved relative intensities only in some cases.

intensity variations within vibrational modes. This had the anticipated gross effect of further broadening the predicted spectra (Fig. 3) to within ~5 cm$^{-1}$ of experiment. Characteristic spectral features were more evident using this approach, such as the doublet observed in cytidine's experimental spectrum (1200–1300 cm$^{-1}$) and the adenosine triplet (1300–1400 cm$^{-1}$), although relative peak intensities remained poorly reproduced across all four nucleosides. For example, the predicted singlet at ~800 cm$^{-1}$ in uridine had an exaggerated intensity (1.00) relative to experiment (0.36). Spectral calculations that modeled the 10 closest water molecules at the B3LYP DFT quantum mechanical level rather than the classical level very marginally improved the predictions (Supplementary Fig. 12), and calculations using an MD ensemble from the generalised AMBER force-field (GAFF) instead of OPLS led to an improvement in predicted spectra for purines, but gave a worse prediction for pyrimidines (Supplementary Fig. 13). Thus, while inclusion of conformer exchange provided a critical step towards a closer agreement with experiment, predicted intensities remained unreliable. Moreover, using this method, applicability to biological molecules was again limited by computational scaling with respect to system size.

**Raman spectral prediction from ab initio molecular dynamics.** In order to address the issue of scalability while also including local dynamics, ab initio molecular dynamics (AIMD) simulations were employed as they afford feasible linear scaling with system size[36]. Specifically, an AIMD simulation of each nucleoside was started from a structure with a *syn*/south conformation (Supplementary Fig. 4). Raman spectra were subsequently determined by Fourier-transforming the autocorrelation-function of the molecular polarizability, further improving computational efficiency by removing the need to compute expensive polarizability derivatives[37]. Local conformer dynamics were sampled for the *syn*/south conformer region per nucleoside, resulting in spectral linewidths that were broadened to within close agreement with experiment (Fig. 3). The adenosine triplet (1300–1400 cm$^{-1}$) was more clearly reproduced in the AIMD-based approach, with the central peak at a much greater intensity (0.92), in line with experiment (1.00). The relative intensities of guanosine peaks at ~1300 and ~1500 cm$^{-1}$ (0.81 *cf.* 1.00) were also more accurately reproduced relative to experiment (0.60 *cf.* 0.99). The *syn* conformers were predicted from the MD simulations (Fig. 2) to be highly populated relative to *anti* for the purines, so it is

unsurprising that AIMD sampling around the *syn* region led to improved experimental agreement. On the other hand, as *anti* conformers were predicted to be of a much higher relative population in pyrimidines, sampling solely around the *syn* regions was expected to lead to poorer experimental agreement. Indeed, the intense doublet in cytidine (~1200–1300 cm$^{-1}$) observed experimentally was not predicted at all, and the intensity of the uridine peak at ~800 cm$^{-1}$ was over-exaggerated (0.96, experimentally 0.36). Therefore, whilst AIMD-simulation of the *syn*/south conformer leads to improvements in terms of linewidth and relative intensities of some purine peaks, for pyrimidines the poor prediction of intensities of key bands confirmed the need to take account of further conformers.

Additional AIMD simulations were carried out to determine whether spectra predicted from local sampling around different conformer regions manifested significant changes to peak intensities. For purines, additional AIMD simulations were carried out in the *syn*/north region only, as it was deemed that the *anti* conformer regions were of too-low a population to contribute significantly overall. For pyrimidines, additional AIMD simulations were performed in *syn*/north and *anti*/south regions (Supplementary Figs. 5 and 6). It was not necessary to perform separate *anti*/north simulations as facile interchange between *anti*/south and *anti*/north conformers was observed, resulting in *anti*/north–south AIMD simulations that contained a mix of both conformers. Raman sub-spectra were subsequently produced for each significantly-populated region using each of these simulated trajectories, as described above (Supplementary Fig. 7). The main differences between sub-spectra (representing each significantly-populated region) were the relative intensities of characteristic peaks. For example, in the *syn*/south sub-spectrum of adenosine, the peak at ~740 cm$^{-1}$ was predicted to have the highest relative intensity (1.00), whereas the same peak in the *syn*/north conformer was much less intense (0.51). Experimentally, this peak was between both predictions (0.64), suggesting contributions from both sub-spectra to the Raman spectrum. Furthermore, in *syn*/north the intensities of the three triplet peaks at ~1300–1400 cm$^{-1}$ (0.78, 1.00, 0.37) were much more accurately reproduced than for *syn*/south (0.49, 0.92, 0.55), relative to experiment (0.71, 1.00, 0.36). For *syn*/south of cytidine, both peaks of the intense, broad doublet (1200–1300 cm$^{-1}$) were predicted to have a very low intensity, indiscernible from spectral noise, whereas in *syn*/north the peak at ~1300 cm$^{-1}$ was recovered, and the peak at ~1250 cm$^{-1}$ was recovered in the *anti*/north–south sub-spectrum. These results exemplify how an overall Raman spectrum comprises spectral features of individual conformer sub-spectra, with substantially different relative intensities. As a consequence, an overall Raman spectrum is a reflection of the relative abundance of each conformer in solution. It was further presupposed that the effect of conformational exchange on Raman spectra could be modeled by mixing and weighting sub-spectra corresponding to significant populations. Moreover, the dramatic intensity changes of characteristic peaks between major conformer sub-spectra provides a theoretical framework for developing empirical rules to rapidly determine conformer populations in more complex molecules.

Sub-spectra for each significant population were recombined using their calculated classical MD weights, which resulted in a substantial *syn* bias for purines and *anti* bias for pyrimidines (Table 1). Additionally, spectra were re-combined by optimizing sub-spectral weights to maximize experimental agreement, thereby allowing experiment to dictate conformer populations (Table 1). Both combination methods led to significantly improved experimental agreement (Fig. 4). For example, inclusion of the *anti* conformers in the cytidine spectra led to the prediction of a strong peak in the 1200–1300 cm$^{-1}$ region. Such a

**Table 1 Relative weighting of theoretical Raman spectra from each conformational region determined from AIMD local sampling. Two differing spectral combination approaches are contrasted in order to reproduce experimental Raman spectra: (A) relative conformer population weights derived from classical MD simulations, and (B) optimization of relative conformer population weights from experiment.**

| Nucleoside | Syn/south | Syn/north | | Anti/north–south | |
| --- | --- | --- | --- | --- | --- |
| | A (%) | B (%) | A (%) | B (%) | A (%) | B (%) |
| Adenosine | 53 | 67 | 31 | 33 | – | – |
| Guanosine | 29 | 55 | 55 | 45 | – | – |
| Cytidine | 4 | 0 | <1 | 18 | 96 | 82 |
| Uridine | 4 | 0 | <1 | 0 | 96 | 100 |

peak was not present in the *syn* spectra. Similarly, in the uridine spectrum, the relative intensities of the peaks at ~800 and ~1250 cm$^{-1}$ were only adequately recovered by heavily weighting spectra towards the *anti* conformers. Both classical MD and optimizing to experiment resulted in strong *anti*-dominance for pyrimidines, with very little difference between spectra computed from each combination method. Likewise for adenosine, differences between each spectral combination method were minimal, with both performing well. This observation suggests that the initial classical MD simulation weights were likely highly representative of solution phase conformer populations. However, for guanosine, weighting optimization to experiment led to improved relative intensities of peaks at ~1200 and ~1300 cm$^{-1}$ when compared to the classical MD predictions. This translates to a likely higher solution population of *syn*/south conformers than predicted by classical MD. To corroborate the *syn*-dominance in purines and *anti*-dominance in pyrimidines, the lowest energy conformer in each conformational region was determined (Supplementary Table 2). For the purines, the lowest energy conformer was in the *syn* region, whereas for pyrimidines, the lowest energy conformer was in the *anti* region (Supplementary Figs. 8–11). This reinforces the prediction that, in solution, *syn* conformers dominate for purines, whereas *anti* conformers dominate for pyrimidines.

**Conformational population validation.** Previous NMR measurements on ribonucleosides[38,39] demonstrated no strong preference for either ring pucker (south/north) in pyrimidines, but a definite bias towards the *anti*-orientation of the base was found and simulations using a specially-parameterized OPLS force-field, as described here, substantiate these conclusions. Similarly, for purines, no strong preference for either ring pucker (south/north) was observed, and while values for base orientation have not been determined by NMR due to aggregation, the OPLS force-field predicts a strong bias towards *syn*-orientation of the base. These previous conclusions are in excellent agreement with our results.

In conclusion, the use of multiple ab initio MD simulations to determine sub-spectra, which are then optimized to experimental Raman spectra, provides an independent measure of ribonucleoside solution populations and is computationally scalable beyond the realm of solution-phase NMR. Our framework thus provides a novel way to deconvolute a Raman spectrum into its constituent conformers in populations that agree with independent NMR measurements. Furthermore, as characteristic Raman bands manifest significant intensity differences between conformers, the opportunity exists to extract empirical rules for the estimation

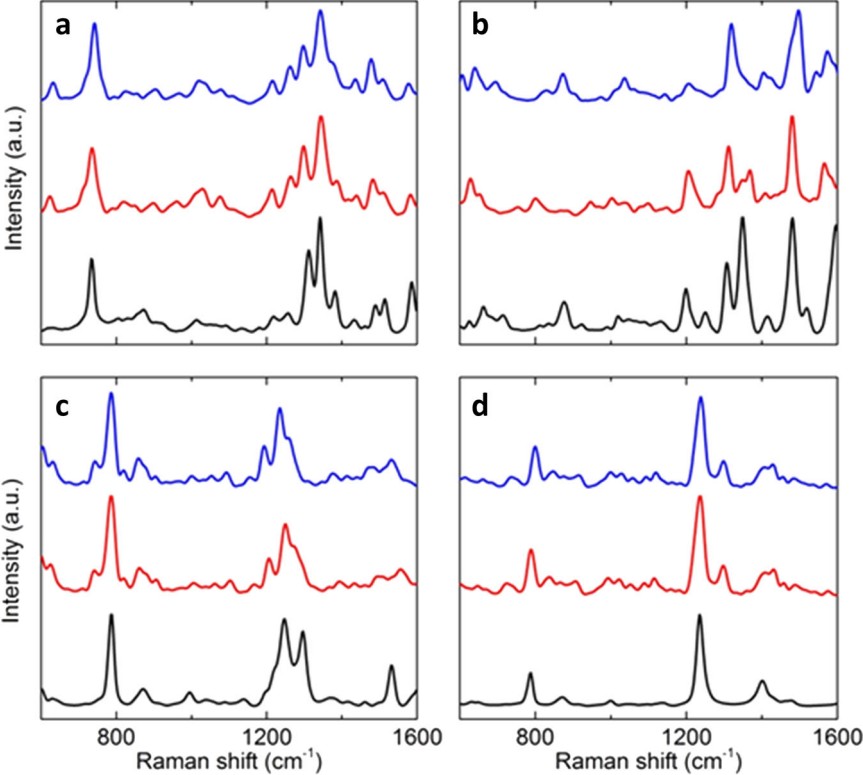

**Fig. 4 Comparison of experiment with AIMD-based theoretical Raman spectra. a** Adenosine, **b** guanosine, **c** cytidine, and **d** uridine. The experimental Raman spectra are shown at the bottom of each panel (black). The two AIMD-based theoretical Raman spectra (plotted above experiment) each represent a combination of spectra calculated from specific conformer regions and include their local dynamics. For the purine nucleosides, AIMD sampling was carried out in the *syn*/south and *syn*/north regions, and the two resulting spectra were combined. For the pyrimidine nucleosides, AIMD sampling was carried out in the *syn*/south, *syn*/north and *anti*/north–south regions. Spectra were combined by weighting each individual spectrum (Supplementary Fig. 7), with two approaches employed to determine spectral weights. Firstly, the classical MD weights of each conformational region were used (blue), and secondly, weights were optimized so that maximum agreement with experiment was achieved (red) (Table 1).

of conformer distributions without recourse to calculation. For example, a *syn*/south adenosine conformer has an intensity ratio of 1.00:0.92 between the intense singlet at ~740 cm$^{-1}$ and the central peak of the triplet at ~1300–1400 cm$^{-1}$, whereas this ratio is 0.69:1.00 in the *syn*/north conformer. Intermediate values for this ratio allow the relative populations of each conformer to be determined and similar associations can be inferred for the other nucleosides. It could be envisaged that in favorable situations and when the sequence of nucleotides is known, empirical rules such as these may be extended to allow for rapid deconvolution of conformational exchange in complex RNA. While the combination of classical and multiple ab initio molecular dynamics simulations with experimental Raman spectra enable a novel route to determine relative populations of RNA nucleosides in solution, it is anticipated that the extension to DNA would be straightforward. Moreover, the framework described here is not restricted to nucleosides and could now be attempted for a wide range of subunits from other important molecules, such as proteins, carbohydrates, and even industrial polymers. In concert with the computational scalability of our framework, this could confer Raman spectroscopy with an ability to explore the conformational exchange within biological subunits more widely. In doing so, spectral complexity can be deciphered and specific bands assigned to specific conformers, overcoming the previous limitations of Raman and opening it up to complement other experimental techniques. This technological advance can lead to new chemical and biological insight, provide a much needed new tool in analytical chemistry, and aid in the manufacture of molecules, such as therapeutics.

## Methods

**Raman spectral prediction from classical molecular dynamics**. For each nucleoside, 5 μs classical molecular dynamics (MD) simulations were performed in water and coordinates were saved every 10 ps for later analysis. Ensembles of 1000 conformations were obtained by uniform sampling of every 50 coordinate sets. Each nucleoside was solvated in a TIP3P[40] water box using the Amber12 tool tleap[41]. Simulations were performed using the OPLS-AA/M force field for RNA[42]. Energy minimization was carried out using the L-BFGS minimization algorithm until the gradient of the internal energy per step was less than 1 kcal mol$^{-1}$. The system was brought to 298 K immediately by suitable scaling of the atomic velocities and then an initial equilibration simulation step in the NPT ensemble for 20 ns was performed using a Monte Carlo barostat (1 atm) with volume changes at 1 ns intervals, coupled to a Langevin thermostat (298 K) with a damping factor of 0.1 ps$^{-1}$. Subsequently, the equilibrium volume was calculated and fixed (45–50 nm$^3$ in all cases) and the simulations were continued in the NVT ensemble for 5 μs. Hydrogen atoms were constrained using M-SHAKE[43], electrostatic interactions were calculated via cubic periodic boundary conditions, electrostatic and van der Waals interactions were truncated (9 Å), and the integration time-step was 4 fs with hydrogen mass (4 a.m.u.) repartitioning[44]. All MD simulations used the OpenMM[45], but with a modification to the Lennard-Jones potential to account for the geometric mixing rule of OPLS-AA/M. A comparison of OPLS-AA/M with other force fields is presented in Supplementary Table 3.

Geometry optimizations were carried out using the ONIOM quantum mechanics/molecular mechanics (QM/MM) implementation in Gaussian09[46]. For each frame the water molecules within 10 Å radius of the nucleoside were kept, and structures were considered optimized when the forces on all nuclei fell below 10$^{-6}$ Hartree/Bohr. Nucleosides were treated with density functional theory (DFT) using the B3LYP functional[47–50], combined with the 6-31+G(d) Pople basis set, and water molecules were treated at the MM level using the Universal force field of Rappé et al.[51] implemented in Gaussian09. No further implicit solvent model was used, as several shells of explicit water were present. It was confirmed that minimization maintained the conformational diversity present in the MD simulations (Supplementary Table 4). Frequencies and activities of Raman-active vibrational modes were determined within the double harmonic approximation using Gaussian09. Raman activities were then converted into intensities by

accounting for excitation laser wavelength (532 nm) and temperature (293 K), using standard methods[52]. These intensities were then normalized for comparison to normalized experimental spectra. Lorentzian broadening to 5 cm$^{-1}$ at full-width half-peak-height, as well as frequency scaling, was performed with an in-house program (see Code availability).

### Raman spectral predictions from ab initio molecular dynamics.
Solvated nucleosides were equilibrated for 2.5 ps followed by 17.5 ps production runs using AIMD, and electron densities were saved every 2.5 fs in the discretized *Gaussian CUBE* format with a stride of 2. The QM subsystem consisted of the single nucleoside, enclosed in a non-periodic cubic box (20.0 Å), in contact with classical water molecules. Global cubic simulation cells were periodic cubic boxes (A = 24.25 Å, U = 23.18 Å, C = 23.43 Å). The QM subsystem used the dispersion-corrected BLYP-D3 functional[48,53,54] and the hybrid Gaussian and planewave (GPW) method[55], core electrons were described with Goedecker–Teter–Hutter (GTH) pseudopotentials[56–58], valence states with the Gaussian-type MOLOPT62 DZVP-MOLOPT-SR-GTH basis set of double-zeta polarized quality from the CP2K package, with a plane-wave cut-off of 280 Ry, and the MM system used identical AMBER parameters to classical MD described above. The equations of motion (classical and Born–Oppenheimer), were integrated using a timestep of 0.5 fs, 300 K using the Nosé thermostat[59–61] (timecon 100 fs) For each timestep, the electronic structure was explicitly quenched to a tolerance of 10$^{-6}$ Hartree. All AIMD simulations were performed with the QUICKSTEP program[36] from the CP2K suite[62]. The Raman spectra were directly determined from the Fourier transform of the autocorrelation function of the molecular polarizability, and were normalized[37]. A positive electric field (5 × 10$^{-4}$ a.u.) was applied along each of the three independent directions of space to get all polarizability tensor components. Numerical integration of the electron density within Voronoi cells used TRAVIS[63].

### Combining ab initio molecular dynamics-derived spectra of individual conformers.
AIMD-derived spectra of multiple conformers of individual nucleosides were combined into a global spectrum for each nucleoside, which was then fitted to experiment. Specifically, the multiple AIMD-derived spectra for each nucleoside were fitted to the corresponding experimental spectrum of that given nucleoside using a statistical procedure, which optimized: (i) the contribution of each spectrum, (ii) the frequency rescaling of every contribution, and (iii) a global frequency shift.

In digital format, spectral data were stored as a two-column array, containing a grid of wavenumbers and the corresponding intensities. Since experimental and predicted spectra often present disjoint wavenumbers, the first step of this process was to obtain intensities on a common grid of $N_{grid}$ wavenumbers for all spectra ($x^{grid,i}$). After normalization, each spectrum was fitted using cubic splines, obtaining a function that describes spectral intensity in terms of the wavenumber. Introducing the rescaling factor of the $j^{th}$ spectrum $r_j$, and the global shift factor $s$, this function has the form of Eq. (1), where $y_i^{conf,j}$ is the intensity prediction for the $i^{th}$ grid point and the $j^{th}$ conformer.

$$y_i^{conf,j}\left(r_j, s\right) = f^{fit,j}\left(r_j x^{grid,i} + s\right) \text{ for } 1 \le i \le N_{grid} \qquad (1)$$

Here, $f^{fit,j}$ is the sum of cubic splines obtained in the previous step, and $y_i^{exp}$ is the experimental intensity at the $i^{th}$ grid point. The parameters mentioned in Eq. (1), as well as the weights ($w_j$), were determined with a modified least-squares regression method, with a loss function that includes corrections to minimize overfitting, Eq. (2). The weights can be interpreted as relative populations of different conformers.

$$L(\mathbf{w}, \mathbf{r}, \mathbf{s}) = \sum_{i=1}^{N_{grid}} \left\{ y_i^{exp} - \left[ \sum_{j=1}^{N_{conf}} w_j y_i^{conf,j} \right] \right\}^2 + L_{weight}(\mathbf{w}) + L_{scaling}(\mathbf{r}) + L_{shift}(s)$$

$$\text{where} \quad L_{weight}(\mathbf{w}) = c_{weight} \left[ \sum_{j=1}^{N_{conf}} w_j^2 - 1 \right]^2$$

$$L_{scaling}(\mathbf{r}) = c_{rescaling} \sum_{j=1}^{N_{conf}} \left( \left| r_j - 1 \right| - r_{eq} \right)^2$$

$$L_{shift}(s) = \begin{cases} c_{shift} & s < s_{min}, s > s_{max} \\ 0 & s_{min} \le s \le s_{max} \end{cases}$$

$$(2)$$

The first term on the right-hand side of Eq. (2) is the classical least-squares loss function that attempts to minimize the error in the predicted spectrum. The second term ensures that the sum of the weights adds up to 1, which is paramount for their interpretation as relative populations. The third term restricts the rescaling factor to lie close to an equilibrium value of ±5%, and the fourth term ensures that the global shift is constrained.

The choice of parameters has a key influence on the effect of this procedure. In this study, the following collection of parameters were used: $c_{weight} = 1.0 \times 10^8$, $c_{rescaling} = 1.0 \times 10^2$, $c_{shift} = 5.0 \times 10^8$, $r_{eq} = 0.05$, $s_{min} = 0.0$ cm$^{-1}$, and $s_{max} = 200.0$ cm$^{-1}$. The least-squares loss function generally takes values in the ~10$^2$ order of magnitude, and therefore we selected penalties that would be loose or tight depending on the violation of the conditions. For example, the large penalty in the weights ensures that they always add up to 1 with significant precision. The

Broyden–Fletcher–Goldfarb–Shanno (BFGS) algorithm was then used to find a minimum of the loss function. In order to find a point close to, or equal to, the global minimum, the optimization procedure was repeated 50 times with different random initial values, keeping the best result.

### Experimental Raman spectra.
The four ribonucleoside samples with purity >99% (Sigma-Aldrich Company Ltd., Gillingham, Dorset, UK) were dissolved in Milli-Q water (Merck KGaA, Darmstadt, Germany). Uridine and cytidine were dissolved at 50 mg/mL, and adenosine and guanosine were made up as saturated solutions to maximize their concentrations. Pyrimidines were self-buffering between pH 7–8 and were not further pH adjusted. Adenosine and guanosine were measured to have pH values of 6.9 and 6.7 at their saturated concentrations, respectively. Raman spectra were acquired using a ChiralRaman spectrometer (BioTools Inc., Jupiter, FL, USA) with a 532 nm laser of power 1.2 W. Samples of 200 μL were placed in rectangular quartz cells coated with magnesium fluoride (Starna Scientific Ltd., Hainault, Essex, UK), sealed to avoid evaporation, and held at a constant temperature (293 K). The final spectra were accumulated over 1 h, and directly prior to each sample acquisition, a spectrum of pure water was obtained under identical conditions. The pure water background was subtracted from the raw sample spectra and the result was further baselined using an asymmetric least squares smoothing approach (third-order polynomial)[64]. Each sample had a sample and laser stabilization of 1 h, at which point it was confirmed that the spectral intensity was constant and thus no sample degradation was occurring nor interference due to undissolved sample. Raman spectra were then recorded over 600–1600 cm$^{-1}$ with a resolution of 3 cm$^{-1}$, using laser illumination periods of 1.25 s. Spectra were truncated at 1600 cm$^{-1}$ as beyond this the water background made baseline correction unreliable, even using the above methods.

## Data availability
The main data supporting the findings of this study are included in the paper and its Supplementary Information files. Additional raw data are available from the corresponding author on reasonable request.

## Code availability
Lorentzian broadening of theoretically predicted Raman spectra, as well as frequency scaling, was performed with an in-house program. The source software code files for performing these calculations are available as Supplementary Data 1 and 2. Two versions are available: one that allows linewidths to be optimized to the Raman spectrum and one that fixes them to 5 cm$^{-1}$ at full-width half-peak-height (as used in Fig. 3). All parameters required are included in the program files.

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

## Acknowledgements

These studies were supported by the UK Research and Innovation Grants (BB/M021637, EP/M507490/1, and EP/K005472).

## Author contributions

A.L.W., A.J.D., P.L.A.P., J.P.W., and A.A. conceived the research. A.A. carried out the classical MD simulations. A.L.W. and C.O. performed all DFT calculations and AIMD simulations. A.L.W. and A.A. carried out the theoretical Raman spectral calculations. S.E.D. performed the experimental Raman spectral determination for each nucleoside. A.L.W. and A.A. wrote the manuscript with input from all authors.

## Competing interests

The authors declare no competing interests.
