## [Peer Review File · Communications Chemistry]

Reviewers' comments:

Reviewer #1 (Remarks to the Author):

The authors present a combined theoretical and experimental study of Raman ribonucleoside spectra. The authors compare different approaches to the calculation of the spectra in order to unveil the origin of the spectroscopic signal for these molecules. The manuscript presents interesting results, however some important changes are required before it can be considered suitable for publication.

1. Throughout the manuscript the authors boldly state that their method can be applicable to polynucleotide chains, far beyond the simple monomers they consider. I believe this conclusion to be far too hasty and in need of being demonstrated. This is for several reasons.

First, while linear scaling of AIMD can be achieved, the complexity of the conformational landscape for a larger chain may grow very quickly, as can the MD time required to explore it, as well as the number of subsequent AIMD simulations.

Secondly, this poses a problem in the decomposition of the resulting spectrum whose information content is almost constant, independent of the size of the system, while the number of conformations with significant contributions may increase significantly, preventing a clear conformational decomposition from being achieved at all.

Until definite proof is provided, the authors should abstain from making such bold claims about the applicability of their method to larger systems, and the manuscript should be rewritten in the affected parts to point out that the present work solely applies to single monomeric nucleosides.

Note that similar approaches employed in the decomposition of, for instance, circular dichroism spectra of proteins in solution, can only produce rough estimates of alpha helix vs. beta sheet and other simple conformational distributions, and nothing more.

2. From the results presented in Table 1 the authors discuss the failure of the MD dynamics to reproduce the conformational distribution of the systems. This is very troublesome as, in some cases such as for cytidine, the MD predicts only 6% anti conformation while spectral decomposition would suggest 100% anti. If the MD is so spectacularly wrong, would AIMD based on structures obtained from that very same dynamics not inherit some bias? In their discussion of previous NMR studies the authors also acknowledge the existence of specially parametrized force-fields, then why not try those for their own work?

3. Because of the previous point, the success of the method over competing approaches in faithfully reproducing experimental results largely rests on the experimental decomposition into the various conformational contributions. However, a fair comparison with other methods, such as the simple spectra obtained from B3LYP optimized minima harmonic frequency calculations requires the very same decomposition to be applied as well. Otherwise, the comparison is highly unfair because those methods would suffer from a bad MD, through no fault of the minimum-energy QM approach itself. The authors should therefore calculate spectra using DFT for the nucleosides in the anti and syn/north conformations, and perform the exact same decomposition of the experimental spectra using the same procedure in order to show that the AIMD results truly beat competing approaches.

Reviewer #2 (Remarks to the Author):

The manuscript describes a detailed computational analysis of possible conformations, which can be adopted by four ribonucleosides, and their corresponding calculated Raman spectra. The authors employ different computational approaches for reproducing Raman spectra, which include DFT-minimized single conformers, classical MD, and ab initio MD. They observed quick interchange between different sugar conformations and pointed at the importance to account for this interchange to obtain agreement with the experimentally measured Raman spectra. The authors

determined the ratios of different conformers providing the best agreement with the experiment. The manuscript is interesting, clearly written and the conclusions are generally supported by the presented data. The described procedure could be used for estimation of different conformer contributions to Raman spectra also for other molecules. However, I would like to draw the author's attention to the latest developments in the field (detailed below), which could be incorporated in their approach and could provide better results for their future research in this direction. I also have a number of points given below, which should be clarified before the manuscript can be published.

1. p.2, line 15, should be "isotopic" instead of "isotropic".
2. p.2, lines 19-20. "...helical nucleotides... Non-helical nucleotides...". Nucleotides are single units consisting of a base, a sugar and a phosphate group and cannot be helical or non-helical. Should be replaced by "polynucleotides" or "oligonucleotides" in this sentence.
3. p.2, line 21. The authors mentioned importance of the sugar interconversion for A-Z RNA and B-Z DNA conformations. However, they did not mention the most important one, B-A DNA conformational change, for which the sugar interconversion plays a crucial role.
4. p.2, line 28. Strong water absorption in infrared is easily overcome by substituting H₂O with D₂O or by using short path length, therefore infrared is as valuable as Raman in solution studies of RNA and DNA.
5. p.2, last 5 lines, and p.3, first 2 lines. The statements provided by the authors are based on relatively old references (7-10 years old, refs. 17-20). There has been a significant progress in the field since then. Most of the problems mentioned by the authors in Introduction have been recently tackled by groups of P.Bour, C. Cappelli and others. For example, the system size accessible for DFT can now be up to 1000 - 10 000 atoms using the Cartesian tensor transfer (CCT) fragmentation technique (Phys. Chem. Chem. Phys. 2018, 20, 4926; J. Am. Chem. Soc. 2011, 133, 15055; Chirality 2010, 22, E96; J. Phys. Chem. Lett. 2015, 6, 3314). Inclusion of conformational exchange as well as accounting for explicit solvent effect and its dynamics can be done by a combined MD/DFT method and similar approaches (e.g., J. Chem. Theory Comput. 2013, 9, 1641; J. Phys. Chem. B 2015, 119, 10682; J. Am. Chem. Soc. 2011, 133, 15055; Chem. Theory Comput. 2017, 13, 4421; Theor. Chem. Acc. 2012, 131, 1201). The authors should update their reference list and their Introduction including these important for the field studies and other relevant new works.
6. p.4, end of the 1st paragraph. The authors state that ribose interconversion occurred on a timescale faster than they sampled the MD trajectory. Why not to sample with higher frequency at least a part of the trajectory to obtain more precise data?
7. p.4, lines 3-4 of the 2nd paragraph and p.5, 3rd line from top. The authors say that after the MD simulations each of the nucleosides was quantum-mechanically energy-minimized. The authors are right, minimization is needed to perform meaningful DFT frequency calculations. However, such a minimization might lead to deviations from the MD-derived structures; eventually all the MD snapshots might collapse to a single structure with lowest energy. This will undermine the whole MD step. Did the authors check whether the quantum-mechanically minimized structures deviated substantially from the initial MD geometries? In order to prevent such a geometry deviation during DFT calculations, the authors might use normal mode optimization (NMO) method developed some time ago in the group of P.Bour but still not very well-known (J. Chem. Phys. 2002, 117, 4126; J. Phys. Chem. B 2015, 119, 10682; J. Am. Chem. Soc. 2011, 133, 15055). The method allows to preserve the overall MD structure while optimizing bond length and angles needed for correct vibrational spectra calculations.
8. p.4, second part of the 2nd paragraph and p.5, 1st paragraph. The authors attribute the highly-

resolved spectra with narrow bands obtained for a single conformer (and their bad agreement with the experiment) to omitted effect of conformational exchange. While this certainly plays a role in natural broadening of the spectral lines, major effect also comes from the solute interaction with water and from water dynamics. This is demonstrated in some of the references I specified above, the authors could look into those examples for more details. Did the authors consider effect of explicit water in their calculations? Did they perform dynamic averaging of the water hydration shell around the solute in their averaging procedure for generating Raman spectra described in the 1st paragraph on page 5? This should be clearly discussed in the text.

9. p.5, last sentence of the 1st paragraph. This statement is not quite correct and is outdated, since, as I already mentioned above, application of CCT method (or other similar approaches) lifts restrictions on the system size up to 1000 – 10 000 atoms, which is sufficient even for relatively large macromolecules.

10. p.6, second part of the 2nd paragraph. The authors talk about "accurate reproduction of experiment", however I do not see accurate agreement between the calculations and the experiment. The authors should restrain from using such phrases, as it can be misleading. Probably it's better to use "better reproduction" or "more accurate reproduction".

11. p.7, 1st paragraph. The authors talk about intramolecular hydrogen bond stabilizing anti/south conformation in pyrimidines and resulting in higher population of this conformation with respect to purines. However, according to fig. 7 - 10 of SI, a similar hydrogen bond is also present in anti/south conformation of purines. In that case why the anti/south population in purines is 0%? How can this be explained?

12. p.7, line 13 from bottom. In case of "complex RNA macromolecules", contributions from all four nucleotides must be considered. Since in general case the sequence of RNA might be unknown, it could be very difficult or even impossible to use this approach for "rapid deconvolution of conformational exchange". This should be considered and discussed in the text. Overall, many statements in the past paragraph on page 7 are very simplistic and vague.

13. p.8, line 4. Why was the protein-optimized forcefield (ff14SB) used for nucleosides? Much more suitable would be GAFF forcefield from Amber package, not optimized for any particular set of molecules. Even using DNA/RNA forcefields (like parm99 and its later modifications) would be not optimal, since they would force the structures to the geometries observed in the macromolecules and will not be relevant to single nucleosides. On the other hand, the experiments are related to single nucleosides, which could eventually result in larger discrepancies between the computed and experimental spectra.

14. p.8, line 5. Why only 50 steps of minimization were done? Normally 1500-2500 steps are performed. How was the heating done? For how long? This is not mentioned.

15. p.8, 2nd paragraph. Was dispersion or PCM used in DFT calculations? Usually PCM applied on top of explicit water molecules improves the results.

16. p.9, experimental section. How could the authors dissolve guanosine at concentration 50 mg/mL and adenosine at 12.5 mg/mL? Solubility of guanosine in water is below 1 mg/mL and solubility of adenosine is around 5 mg/mL.

17. figs.1 and 2. The color reproduction in the diagrams does not correspond to the description, e.g. there is violet color instead of pink for c3'-endo in fig.1, yellow instead of blue for anti/south and blue instead of cyan for anti/north in fig.2. Please check this.

18. fig.4. Why were the spectra truncated at 1600cm⁻¹? There are still some bands above that wavenumber.

19. Table 1 of SI. What does "nu-omega Base (C-C and C-O)" vibration mean for U at 994 cm⁻¹? Does it mean that both stretching and wagging vibrations contribute to this band? In that case they should be written separately to avoid confusion. Also, according to the convention, letter "nu" which is used for denoting stretching vibrations should be small (not capital) and "sym" or "asym" should be with all small letters (first S or A is not capital) and indicated as subscript of "nu". Also, notation "(2' 3' C-C)" is not very clear and should be changed to "(C2'-C3')" to follow the general convention.

20. Fig. 6 of SI. The labels mentioned in the figure caption and in the figure do not correspond. The figure panels should be marked a), b), c) and d), while the green spectra should be marked i), ii) and iii) from top to bottom in each panel.

21. Figs. 7-10 of SI. It would be useful to put labels with calculated energy for each conformer.

Specific responses to the referees' comments.

Referee 1.

Q1. Throughout the manuscript the authors boldly state that their method can be applicable to polynucleotide chains, far beyond the simple monomers they consider. I believe this conclusion to be far too hasty and in need of being demonstrated. This is for several reasons.

First, while linear scaling of AIMD can be achieved, the complexity of the conformational landscape for a larger chain may grow very quickly, as can the MD time required to explore it, as well as the number of subsequent AIMD simulations.

Secondly, this poses a problem in the decomposition of the resulting spectrum whose information content is almost constant, independent of the size of the system, while the number of conformations with significant contributions may increase significantly, preventing a clear conformational decomposition from being achieved at all.

Until definite proof is provided, the authors should abstain from making such bold claims about the applicability of their method to larger systems, and the manuscript should be rewritten in the affected parts to point out that the present work solely applies to single monomeric nucleosides. Note that similar approaches employed in the decomposition of, for instance, circular dichroism spectra of proteins in solution, can only produce rough estimates of alpha helix vs. beta sheet and other simple conformational distributions, and nothing more.

A1. We have removed references to RNA macromolecules throughout the text. For example in the abstract we have written, "providing a novel route to explore conformational exchange in RNA subunits." We note that towards the end of the introduction we now clearly state, "While we demonstrate this for the first time on RNA ribonucleoside building blocks, it may have applicability to a wide range of other molecules." We believe that we have now made the immediate applicability of our method clear. We hope that the reviewer is in agreement and we thank them for their comments.

Q2. From the results presented in Table 1 the authors discuss the failure of the MD dynamics to reproduce the conformational distribution of the systems. This is very troublesome as, in some cases such as for cytidine, the MD predicts only 6% anti conformation while spectral decomposition would suggest 100% anti. If the MD is so spectacularly wrong, would AIMD based on structures obtained from that very same dynamics not inherit some bias? In their discussion of previous NMR studies the authors also acknowledge the existence of specially parametrized force-fields, then why not try those for their own work?

A2. We have now rerun the MD calculations with 3 different force-fields, including OPLS and GAFF. The results from all these new calculations can be seen in Supplementary Table 3. The OPLS force-field is a specially-parametrized force-field for RNA and was published very recently (2019). The results from this force field are much more consistent with our Raman results and previous experimental data (as can be seen in updated Figure 2 and Table 1). As a result, we have removed the previous calculations from the main text and moved these data to Supplementary Table 3. It has also necessitated rerunning all of the DFT spectral calculations and a recalculation of Figure 3, which has been fully redrawn in the revision. Also, updates have had to be made to the Supplementary Figures and Tables as a result of the new MD and DFT calculations.

Q3. Because of the previous point, the success of the method over competing approaches in faithfully reproducing experimental results largely rests on the experimental decomposition into the various conformational contributions. However, a fair comparison with other methods, such as the simple

spectra obtained from B3LYP optimized minima harmonic frequency calculations requires the very same decomposition to be applied as well. Otherwise, the comparison is highly unfair because those methods would suffer from a bad MD, through no fault of the minimum-energy QM approach itself. The authors should therefore calculate spectra using DFT for the nucleosides in the anti and syn/north conformations, and perform the exact same decomposition of the experimental spectra using the same procedure in order to show that the AIMD results truly beat competing approaches.

A3. We have extracted the conformers from the MD and performed DFT calculation on them. A new figure has been added to the SI (Supplementary Figure 3) to confirm that these DFT calculations in any of the canonical nucleoside conformers (*anti/south* etc.) are inferior to AIMD. We have added a statement to the text, “Therefore, it is apparent that predictions based on the single most populated conformer (Fig. 3) and any of the other canonical conformers (Supplementary Fig. 3) may be insufficient to interpret and decipher the complex Raman spectra of macromolecular systems, such as RNA.”

Referee 2

Q1. p.2, line 15, should be “isotopic” instead of “isotropic”.

A1. We have made this correction. We thank the reviewer for their attention to detail.

Q2. p.2, lines 19-20. “...helical nucleotides.... Non-helical nucleotides...”. Nucleotides are single units consisting of a base, a sugar and a phosphate group and cannot be helical or non-helical. Should be replaced by “polynucleotides” or “oligonucleotides” in this sentence.

A2. We have made the change from nucleotides to polynucleotides.

Q3. p.2, line 21. The authors mentioned importance of the sugar interconversion for A-Z RNA and B-Z DNA conformations. However, they did not mention the most important one, B-A DNA conformational change, for which the sugar interconversion plays a crucial role.

A3. We have made the change to “Dynamical interconversion is therefore responsible for important macromolecular changes between A and Z RNA and between A, B and Z DNA”

Q4. p.2, line 28. Strong water absorption in infrared is easily overcome by substituting H₂O with D₂O or by using short path length, therefore infrared is as valuable as Raman in solution studies of RNA and DNA.

A4. We have changed these sentences to “Both infrared and Raman spectroscopy are able to study solution phase RNA samples, without restrictions on molecular size, and potentially provide an inexpensive, rapid route to the elucidation of RNA conformation.” We trust that the reviewer is happy with this change.

Q5. p.2, last 5 lines, and p.3, first 2 lines. The statements provided by the authors are based on relatively old references (7-10 years old, refs. 17-20). There has been a significant progress in the field since then. Most of the problems mentioned by the authors in Introduction have been recently tackled by groups of P.Bour, C. Cappelli and others. For example, the system size accessible for DFT can now be up to 1000 - 10 000 atoms using the Cartesian tensor transfer (CCT) fragmentation technique (Phys. Chem. Chem. Phys. 2018, 20, 4926; J. Am. Chem. Soc. 2011, 133, 15055; Chirality 2010, 22, E96; J. Phys. Chem. Lett. 2015, 6, 3314). Inclusion of conformational exchange as well as accounting for explicit solvent effect and its dynamics can be done by a combined MD/DFT method and similar approaches (e.g., J. Chem. Theory Comput. 2013, 9, 1641; J. Phys. Chem. B 2015, 119, 10682; J. Am. Chem. Soc. 2011, 133, 15055; Chem. Theory Comput. 2017, 13, 4421; Theor. Chem. Acc. 2012, 131,1201). The authors should update their reference list and their Introduction including these important for the field studies and other relevant new works.

A5. We have added a statement to this effect, “However, there have been recent successful attempt to simulate on the order of 10^3 - 10^4 atoms using combined classical and quantum methods,” and included the suggested references (e.g., see new references #21-28).

Q6. p.4, end of the 1st paragraph. The authors state that ribose interconversion occurred on a timescale faster than they sampled the MD trajectory. Why not to sample with higher frequency at least a part of the trajectory to obtain more precise data?

A6. The MD sample frequency has been increased by ten times in the current calculations (see below A13) allowing us to estimate the timescales. We have also added the following text, “The exchange rate between conformations was in the range 0.1 - 3.3 ns^{-1} (Supplementary Figs. 1 and 2), putting the mean lifetimes of conformers in the 100s of picoseconds to 10s of nanoseconds range, in agreement with fast exchange observed in previous NMR studies²³. In the pyrimidines anti to syn transitions had the slower rate ($\sim 0.1 \text{ ns}^{-1}$), consistent with the preference for the *anti* conformer, while in purines it was the *syn* to anti transition that had the slower rate ($\sim 0.4 \text{ ns}^{-1}$).”

Q7. p.4, lines 3-4 of the 2nd paragraph and p.5, 3rd line from top. The authors say that after the MD simulations each of the nucleosides was quantum-mechanically energy-minimized. The authors are right, minimization is needed to perform meaningful DFT frequency calculations. However, such a minimization might lead to deviations from the MD-derived structures; eventually all the MD snapshots might collapse to a single structure with lowest energy. This will undermine the whole MD step. Did the authors check whether the quantum-mechanically minimized structures deviated substantially from the initial MD geometries? In order to prevent such a geometry deviation during DFT calculations, the authors might use normal mode optimization (NMO) method developed some time ago in the group of P.Bour but still not very well-known (J. Chem. Phys. 2002, 117, 4126; J. Phys. Chem. B 2015, 119, 10682; J. Am. Chem. Soc. 2011, 133, 15055). The method allows to preserve the overall MD structure while optimizing bond length and angles needed for correct vibrational spectra calculations.

A7. We have added a table to the SI to confirm this (Supplementary Table 4). We have added a sentence to the text, “It was confirmed that minimization maintained the conformational diversity present in the MD simulations (Supplementary Table 4).” on p.9 paragraph 2 of the Methods.

Q8. p.4, second part of the 2nd paragraph and p.5, 1st paragraph. The authors attribute the highly-resolved spectra with narrow bands obtained for a single conformer (and their bad agreement with the experiment) to omitted effect of conformational exchange. While this certainly plays a role in natural broadening of the spectral lines, major effect also comes from the solute interaction with water and from water dynamics. This is demonstrated in some of the references I specified above, the authors could look into those examples for more details. Did the authors consider effect of explicit water in their calculations? Did they perform dynamic averaging of the water hydration shell around the solute in their averaging procedure for generating Raman spectra described in the 1st paragraph on page 5? This should be clearly discussed in the text.

A8. We did use explicit solvent (a shell of ~ 300 water molecules as predicted by MD) in the calculations throughout. We direct the reviewer to the specific sentences where we mention this, “For each frame the water molecules within 10 \AA radius of the nucleoside were kept,” and “water molecules were treated at the MM level using the Universal force field”

Q9. p.5, last sentence of the 1st paragraph. This statement is not quite correct and is outdated, since, as I already mentioned above, application of CCT method (or other similar approaches) lifts restrictions on the system size up to 1000 – 10 000 atoms, which is sufficient even for relatively large macromolecules.

A9. We changed the sentence to, “Moreover, using this method, applicability to macromolecules was again limited by computational scaling with respect to system size.”

Q10. p.6, second part of the 2nd paragraph. The authors talk about “accurate reproduction of experiment”, however I do not see accurate agreement between the calculations and the experiment. The authors should restrain from using such phrases, as it can be misleading. Probably it’s better to use “better reproduction” or “more accurate reproduction”.

A10. We have changed this to “more accurate reproduction.”

Q11. p.7, 1st paragraph. The authors talk about intramolecular hydrogen bond stabilizing anti/south conformation in pyrimidines and resulting in higher population of this conformation with respect to purines. However, according to fig. 7 - 10 of SI, a similar hydrogen bond is also present in anti/south conformation of purines. In that case why the anti/south population in purines is 0%? How can this be explained?

A11. Yes, in retrospect we agree with the reviewer that this is perhaps more complicated and have updated the sentence to “due to the relative stabilizing effects of intramolecular hydrogen bonding,” rather than referring to specific hydrogen bonds.

Q12. p.7, line 13 from bottom. In case of “complex RNA macromolecules”, contributions from all four nucleotides must be considered. Since in general case the sequence of RNA might be unknown, it could be very difficult or even impossible to use this approach for “rapid deconvolution of conformational exchange”. This should be considered and discussed in the text. Overall, many statements in the past paragraph on page 7 are very simplistic and vague.

A12. We have made changes in this section, such as “It could be envisaged that in favourable situations and when the sequence of nucleotides is known, empirical rules such as these may be extended to allow for rapid deconvolution of conformational exchange in complex RNA macromolecules.”

Q13. p.8, line 4. Why was the protein-optimized forcefield (ff14SB) used for nucleosides? Much more suitable would be GAFF forcefield from Amber package, not optimized for any particular set of molecules. Even using DNA/RNA forcefields (like parm99 and its later modifications) would be not optimal, since they would force the structures to the geometries observed in the macromolecules and will not be relevant to single nucleosides. On the other hand, the experiments are related to single nucleosides, which could eventually result in larger discrepancies between the computed and experimental spectra.

A13. We have rerun the calculations with 3 new force-fields. Including OPLS and GAFF. The OPLS force-field is very recent (2019) and is now included in the text. The results from this force field appear to be much more consistent with our results and previous experimental data. We thank the reviewer for pointing this out. The other results have been included in the SI. We have also increased the sampling rate to every 10ps in order to further investigate the rate of conformational exchange. This results in half a million structures per 5 microsecond simulation.

Q14. p.8, line 5. Why only 50 steps of minimization were done? Normally 1500-2500 steps are performed. How was the heating done? For how long? This is not mentioned.

A14. This all depends on the system size in our opinion. The minimization is purely to release any bad contacts and get the gradient of force to an acceptable level to start the simulations. In the new simulations minimization was performed until the gradient of the internal energy per step was less than 1 kcalmol⁻¹. An explicit heating phase was not included. We have included text, “The velocities are randomly set to a distribution based on the target temperature and equilibration performed at the target temperature”. This is now standard practice and no energy instabilities in the simulations were observed.

Q15. p.8, 2nd paragraph. Was dispersion or PCM used in DFT calculations? Usually PCM applied on top of explicit water molecules improves the results.

A15. We included a substantial amount of explicit water (300 water molecules), which is several shells of water and would be no edge effect at this distance. We have added comments about the included water. “For each frame the water molecules within 10 Å radius of the nucleoside were kept.” In this case we did not opt to also include further implicit solvent models. We have also made a comment about that, “No further implicit solvent model was used, as several shells of explicit water were present.”

Q16. p.9, experimental section. How could the authors dissolve guanosine at concentration 50 mg/mL and adenosine at 12.5 mg/mL? Solubility of guanosine in water is below 1 mg/mL and solubility of adenosine is around 5 mg/mL.

A16. The referee is completely correct. Adenosine and guanosine were made up as saturated solutions to maximise their concentrations. The amounts we referred to in the text was the amount added and it is true that most of it would not dissolve and was removed. This has been corrected in the revised manuscript.

Q17. figs.1 and 2. The color reproduction in the diagrams does not correspond to the description, e.g. there is violet color instead of pink for c3'-endo in fig.1, yellow instead of blue for anti/south and blue instead of cyan for anti/north in fig.2. Please check this.

A17. We have checked these and updated as required.

Q18. fig.4. Why were the spectra truncated at 1600cm⁻¹? There are still some bands above that wavenumber.

A18. Spectra were truncated at 1600cm⁻¹ as they are affected by a high water background after this. Baseline correction beyond this was not reliable. We have added, “Spectra were truncated at 1600cm⁻¹ as beyond this the water background made baseline correction unreliable.”

Q19. Table 1 of SI. What does “nju-omega Base (C-C and C-O)” vibration mean for U at 994 cm⁻¹? Does it mean that both stretching and wagging vibrations contribute to this band? In that case they should be written separately to avoid confusion. Also, according to the convention, letter “nju” which is used for denoting stretching vibrations should be small (not capital) and “sym” or “asym” should be with all small letters (first S or A is not capital) and indicated as subscript of “nju”. Also, notation “(2' 3' C-C)” is not very clear and should be changed to “(C2'-C3')” to follow the general convention.

A19. Yes, we agree and we have made these changes to the table.

Q20. Fig. 6 of SI. The labels mentioned in the figure caption and in the figure do not correspond. The figure panels should be marked a), b), c) and d), while the green spectra should be marked i), ii) and iii) from top to bottom in each panel.

A20. We have made these changes.

Q21. Figs. 7-10 of SI. It would be useful to put labels with calculated energy for each conformer.

A21. We have added text to each figure legend, “The energies relative to the syn/south conformer are of x, y, z, and a for syn/south, syn/north, anti/south and anti/north, respectively.” These are now Supplementary Figures 8-11.

Reviewers' comments:

Reviewer #1 (Remarks to the Author):

The authors have satisfactorily amended the manuscript. It can be published in its present form.

Reviewer #2 (Remarks to the Author):

The authors made some improvements to the manuscript, however, many points are still unclear and unconvincing.

1. The authors use large water shell (300 water molecules) in their DFT calculations, however, the water molecules are treated at the MM level. In order to account for the explicit water effect on the spectra, water molecules should be treated at the same DFT level as the solute molecule. It is usually enough to include only the closest 10-20 water molecules to account for the most of the water effect, but they should be dynamically averaged. I believe this would further improve the band width agreement with the experiment.

2. p.5, last paragraph. Why was syn/south conformation used as a starting geometry for all AIMD simulations? According to the results of classical MD in fig. 2, this is the most populated conformer for pyrimidines, but not for purines.

3. It is not quite clear from fig.2 (and fig.3), which classical force field was used for the presented results. I assume these are the results obtained with OPLS force field. This should be clearly stated in the figure captions and in the text.

4. The results obtained for different force fields presented in Table 3 of SI are very different. Particularly strikingly different are the results from GAFF force field. Did the authors do any calculations of the spectra based on the conformer distribution obtained from GAFF MD? It would be interesting to compare the GAFF-calculated spectra with the experimental ones. Ideally, GAFF should provide the most correct geometries for the monomers, not biased by special over-parametrisation towards RNA polymer geometry.

5. p.7, 3rd line from the bottom. The discussion of the importance of the intra-molecular hydrogen bond for stabilization of a particular conformation is again not fully supported by the presented data. All the presented pyrimidine conformers in figs. 10 and 11 of SI have hydrogen bonds. Therefore, the simple statement that presence of the hydrogen bond stabilizes a particular conformation is not enough. Perhaps, the authors could present some comparison of the hydrogen bond geometries in all the conformers (length, angle) and correlate that with the calculated conformer energy. Such data could possibly support the author's statement, otherwise this statement should be removed.

6. p.8, last paragraph. I'm still quite sceptical about author's statements regarding expanding their framework to "complex RNA", particularly based on such a vague estimation as the intensity ratio of the bands in the calculated spectra. Without any definite proofs such statements should be avoided and until these proofs are obtained, only monomers or similar small molecules could be discussed.

7. Experimental part. Why wasn't Raman spectrum of water (solvent) subtracted from the spectra of the samples before doing any baseline correction? This is a general procedure for most of spectroscopic methods. This would easily remove the water band around 1660cm⁻¹ and allow to present the whole spectrum. Furthermore, often this would also automatically take care of the baseline and in many cases no additional baseline correction would be needed, which decreases introduction of any artifacts due to heavy spectra processing.

8. How was the undissolved guanosine and adenosine removed from the Raman samples? Presence of undissolved particles in the sample may result in strong scattering, preventing correct measurements.

9. The samples can indeed self-buffer to pH around 7 at high concentrations, however at low concentrations (as in the case of guanosine and adenosine), the self-buffering capability is low and with standard milliQ water the pH is usually around 5.

10. The laser power used by the authors (1.2W) might be too high, resulting in local heating and

damage of the sample. Did the authors check the stability of the samples? Further irradiating the samples during burning off the fluorescence can contribute to this effect.

11. If the samples are pure enough, there should be no fluorescence and no need to burn it off. Usually the fluorescence in non-fluorescing samples comes from the impurities in the sample and can be avoided by careful sample preparation or changing the sample provider. Also, scattering may result in a similar spectral line distortion. Are the authors sure that their samples were pure and that they indeed had fluorescence and not scattering? If the undissolved particles were present in the sample, they could result in the scattering. The scattering may decrease upon particle precipitation after some time. In this case, no burning off is needed.

12. Table 1 of SI. Some of the abbreviations and notations used by the authors still do not correspond to the generally accepted in spectroscopy convention. Symmetric stretching vibration is usually denoted not as "nju & sym", but as greek non-capital "nju" with latin non-capital "sym" or "s" as a subscript. Asymmetric stretching vibration is denoted as greek non-capital "nju" with latin non-capital "asym" or "as" or "a" as a subscript. Also, the prime notation for atoms in nucleic acids (e.g., C2') is used only for the sugar atoms, but not for the nitrogen base atoms (as, e.g., the authors used for 993cm⁻¹ of C (C5'=C6')). The authors are advised to check some spectroscopic literature or standards for correct abbreviations and nomenclature.

Specific responses to the review comments.

Reviewer #1

We wish to thank the reviewer for previously providing positive and detailed feedback for our manuscript and welcome their opinion that it can be published in its present form.

Reviewer #2

Q1. The authors use large water shell (300 water molecules) in their DFT calculations, however, the water molecules are treated at the MM level. In order to account for the explicit water effect on the spectra, water molecules should be treated at the same DFT level as the solute molecule. It is usually enough to include only the closest 10-20 water molecules to account for the most of the water effect, but they should be dynamically averaged. I believe this would further improve the band width agreement with the experiment.

A1. In order to address this we performed further DFT calculations. In this case we moved the closest 10 water molecules to the nucleoside molecule into the same B3LYP level of theory as the nucleoside, and the other 290 water molecules were kept at the MM level of theory. The resultant spectra were largely unchanged, and importantly, the agreement with experiment showed only a marginal improvement (see Supplementary Figure 12). We have added a note to the text, p.5 paragraph 2, "Spectral calculations that modelled the 10 closest water molecules at the B3LYP DFT quantum mechanical level rather than the classical level very marginally improved the predictions (Supplementary Fig. 12)", and the calculated spectra have been placed in the Supplementary Information.

Q2. p.5, last paragraph. Why was syn/south conformation used as a starting geometry for all AIMD simulations? According to the results of classical MD in fig. 2, this is the most populated conformer for pyrimidines, but not for purines.

A2. All the simulations were started in the same geometry so as not to bias the AIMD simulations. We note that the AIMD simulations moved away from their starting geometries and we are confident that the starting geometries did not affect the conclusions that were drawn from the AIMD analysis.

Q3. It is not quite clear from fig.2 (and fig.3), which classical force field was used for the presented results. I assume these are the results obtained with OPLS force field. This should be clearly stated in the figure captions and in the text.

A3. We decided to use the OPLS-AA/M force field based on a balanced view of all the reviewer comments. OPLS-AA/M was actually the most difficult for us to implement but it is currently the state-of-the-art for these molecules. We have updated the text and figure legends to make this clearer and we thank the reviewer for pointing this out. For example, on p.5 paragraph 2 we included "using aqueous molecular dynamics simulations and a force-field optimized for RNA (OPLS-AA/M)."

Q4. The results obtained for different force fields presented in Table 3 of SI are very different. Particularly strikingly different are the results from GAFF force field. Did the authors do any calculations of the spectra based on the conformer distribution obtained from GAFF MD? It would be interesting to compare the GAFF-calculated spectra with the experimental ones. Ideally, GAFF should provide the most correct geometries for the monomers, not biased by special over-parametrisation towards RNA polymer geometry.

A4. We have repeated the DFT calculations using GAFF and found that it did not uniformly improve the agreement with experiment – the pyrimidines were worse, while the purines performed better. Certainly this was not significant enough to warrant us making major changes to the results. We have included the spectral calculations from GAFF in the Supplementary Information and added text to p.5 paragraph 2 of the manuscript, “and calculations using an MD ensemble from the generalised AMBER force-field (GAFF) instead of OPLS led to an improvement in predicted spectra for purines, but gave a worse prediction for pyrimidines (Supplementary Fig. 13).” While some readers may find this of interest, we trust that the reviewer will appreciate that determining the benefits and deficiencies of different MD force-fields is beyond the scope of our present manuscript.

Q5. p.7, 3rd line from the bottom. The discussion of the importance of the intra-molecular hydrogen bond for stabilization of a particular conformation is again not fully supported by the presented data. All the presented pyrimidine conformers in figs. 10 and 11 of SI have hydrogen bonds. Therefore, the simple statement that presence of the hydrogen bond stabilizes a particular conformation is not enough. Perhaps, the authors could present some comparison of the hydrogen bond geometries in all the conformers (length, angle) and correlate that with the calculated conformer energy. Such data could possibly support the author’s statement, otherwise this statement should be removed.

A5. We have removed the statement in the latest revision.

Q6. p.8, last paragraph. I’m still quite sceptical about author’s statements regarding expanding their framework to “complex RNA”, particularly based on such a vague estimation as the intensity ratio of the bands in the calculated spectra. Without any definite proofs such statements should be avoided and until these proofs are obtained, only monomers or similar small molecules could be discussed.

A6. On this point we disagree with the referee. We firmly believe that it is written in a cautious and future-looking manner that does not misinform readers or claim that we have definitive evidence at this time. While we understand the reviewer’s concerns we believe that the reviewer is taking an overly draconian position in wanting us to remove this part of our discussion and moreover we note that independent reviewer #1 found our form of words to be acceptable.

Q7. Experimental part. Why wasn’t Raman spectrum of water (solvent) subtracted from the spectra of the samples before doing any baseline correction? This is a general procedure for most of spectroscopic methods. This would easily remove the water band around 1660cm⁻¹ and allow to present the whole spectrum. Furthermore, often this would also automatically take care of the baseline and in many cases no additional baseline correction would be needed, which decreases introduction of any artifacts due to heavy spectra processing.

A7. We are sorry for any misunderstanding here. The method that we used was as largely described in citation 64. A spectrum of pure water was recorded directly prior to acquisition of each sample spectrum (at the exact experimental configuration: laser power etc.). The water background was subtracted from the acquired sample spectrum (as the reviewer intimates). The resultant spectrum was then baselined using third order polynomials as described in detail in citation 64 (beyond the scope and space of our

manuscript). We have modified the following text to p.10 paragraph 2, “The final spectra were accumulated over 1 h, and directly prior to each sample acquisition, a spectrum of pure water was obtained under identical conditions. The pure water background was subtracted from the raw sample spectra and the result was further baselined using an asymmetric least squares smoothing approach (third order polynomial).”

Q8. How was the undissolved guanosine and adenosine removed from the Raman samples? Presence of undissolved particles in the sample may result in strong scattering, preventing correct measurements.

A8. Any undissolved guanosine and adenosine were allowed to settle and fall out of the path of the laser. The experimental set up allowed for only the suspended (dissolved) sample to be in the path of the laser, by looking at a small window at only the top of the cuvette where samples were dissolved. We added the following text to p.10 paragraph 2, “Each sample had a sample and laser stabilisation of 1 h, at which point it was confirmed that the spectral intensity was constant and thus neither sample degradation was occurring nor interference due to undissolved sample.”

Q9. The samples can indeed self-buffer to pH around 7 at high concentrations, however at low concentrations (as in the case of guanosine and adenosine), the self-buffering capability is low and with standard milliQ water the pH is usually around 5.

A9. Sample pHs were confirmed using a mini pH probe. Guanosine and adenosine were confirmed to have pHs of 6.7 and 6.9, respectively. We modified the text (p.10 paragraph 2) to “Pyrimidines were self-buffering between pH 7-8 and were not further pH adjusted. Adenosine and guanosine were measured to have pH values of 6.9 and 6.7 at their saturated concentrations, respectively.”

Q10. The laser power used by the authors (1.2W) might be too high, resulting in local heating and damage of the sample. Did the authors check the stability of the samples? Further irradiating the samples during burning off the fluorescence can contribute to this effect.

A10. The Raman spectrum remained stable during the acquisition period (1-hr) and we also have data to show this was also the case over 24hrs. We have modified the text (p.10 paragraph 2) to “Each sample had a sample and laser stabilisation of 1 h, at which point it was confirmed that the spectral intensity was constant and thus neither sample degradation was occurring nor interference due to undissolved sample.”

Q11. If the samples are pure enough, there should be no fluorescence and no need to burn it off. Usually the fluorescence in non-fluorescing samples comes from the impurities in the sample and can be avoided by careful sample preparation or changing the sample provider. Also, scattering may result in a similar spectral line distortion. Are the authors sure that their samples were pure and that they indeed had fluorescence and not scattering? If the undissolved particles were present in the sample, they could result in the scattering. The scattering may decrease upon particle precipitation after some time. In this case, no burning off is needed.

A11. Samples were pure and the Raman spectra peaks correlate (and have been compared) to those found in both poly-RNA samples and samples of longer RNA oligonucleotides. It’s likely that any ‘burn off’ would have come from scattering (rather than fluorescence) – the referee is probably correct. However, we anticipated that there may be scattering of the sample due to the poor solubility and thus ensured that any insoluble material settled out over an hour before data acquisition. We have removed

reference to fluorescence and written the more generic (p.10 paragraph 2), “Each sample had a sample and laser stabilisation of 1 h..”

Q12. Table 1 of SI. Some of the abbreviations and notations used by the authors still do not correspond to the generally accepted in spectroscopy convention. Symmetric stretching vibration is usually denoted not as “ ν & sym”, but as greek non-capital “ ν ” with latin non-capital “sym” or “s” as a subscript. Asymmetric stretching vibration is denoted as greek non-capital “ ν ” with latin non-capital “asym” or “as” or “a” as a subscript. Also, the prime notation for atoms in nucleic acids (e.g., C2’) is used only for the sugar atoms, but not for the nitrogen base atoms (as, e.g., the authors used for 993cm⁻¹ of C (C5’=C6’)). The authors are advised to check some spectroscopic literature or standards for correct abbreviations and nomenclature.

A12. We have tried to make the changes requested by the reviewer regarding the abbreviations and nomenclature. However, we are aware that different groups and scientific papers use different abbreviations and nomenclature. For the avoidance of doubt, we have provided a key to the abbreviations.

REVIEWERS' COMMENTS:

Reviewer #2 (Remarks to the Author):

The authors replied to the questions satisfactory and made the corresponding changes in the text. I noticed only two minor misprints in table S1. Please correct them in the final version of the manuscript.

1. Table S1, U at 657cm⁻¹ (calculated). Make "nju" not capital.
2. Table S1, C at 1612cm⁻¹ (calculated). The authors missed one instance of "nju & asym". Please change it in accordance with all the others.

Specific responses to the review comments.

Reviewer #2

Q1. Table S1, U at 657cm⁻¹ (calculated). Make “nju” not capital Table S1, C at 1612cm⁻¹ (calculated). The authors missed one instance of “nju & asym”. Please change it in accordance with all the others.

A1. We have made these changes in the revised version.